# Design and Implementation of a Wearable Accelerometer-Based Motion/Tilt Sensing Internet of Things Module and Its Application to Bed Fall Prevention

**DOI:** 10.3390/bios11110428

**Published:** 2021-10-29

**Authors:** Wen-Yen Lin, Chien-Hung Chen, Ming-Yih Lee

**Affiliations:** 1Center for Biomedical Engineering, Department of Electrical Engineering, Chang Gung University, Tao-Yuan 33302, Taiwan; 2Division of Cardiology, Department of Internal Medicine, Chang Gung Memorial Hospital, Tao-Yuan 33302, Taiwan; leemiy@mail.cgu.edu.tw; 3Graduate Institute of Biomedical Engineering, Chang Gung University, Tao-Yuan 33302, Taiwan; d0528005@cgu.edu.tw

**Keywords:** accelerometer, wearable sensor, bed-fall prevention, motion and tilt sensing, health-care, Internet-of-Things

## Abstract

Accelerometer-based motion sensing has been extensively applied to fall detection. However, such applications can only detect fall accidents; therefore, a system that can prevent fall accidents is desirable. Bed falls account for more than half of patient falls and are preceded by a clear warning indicator: the patient attempting to get out of bed. This study designed and implemented an Internet of Things module, namely, Bluetooth low-energy-enabled Accelerometer-based Sensing In a Chip-packaging (BASIC) module, with a tilt-sensing algorithm based on the patented low-complexity COordinate Rotation DIgital Computer (CORDIC)-based algorithm for tilt angle conversions. It is applied for detecting the postural changes (from lying down to sitting up) and to protect individuals at a high risk of bed falls by prompting caregivers to take preventive actions and assist individuals trying to get up. This module demonstrates how motion and tilt sensing can be applied to bed fall prevention. The module can be further miniaturized or integrated into a wearable device and commercialized in smart health-care applications for bed fall prevention in hospitals and homes.

## 1. Introduction

Wearable sensors have gained attention in recent decades because they allow for noninvasive, real-time physical, and physiological monitoring, especially in smart care and remote digital medicines [1,2,3]. Wearable devices have been used by various groups of individuals, such as athletes [4], to maximize their health, monitor their training and improve their performance, and purposes such as rehabilitation [5]. Wearable sensors are a part of Internet of Things (IoT) system. An IoT system comprises the following fundamental components: sensors, connectivity, and applications. Thus, a wearable sensor, its connectivity, and its health-care applications form a complete IoT system. In this system, data acquired by the sensor are algorithmically processed for dedicated applications, and users interact with the processed data via graphical user interfaces. Common types of sensing include biochemical sensing [6,7,8]; biological sensing, such as electrocardiography (ECG) [9], electromyography (EMG) [10], and electroencephalography (EEG)-based neural status [11]; and motion sensing [12,13]. Motion sensing systems most commonly include an accelerometer that can detect four of the five types of inertial motions: acceleration, vibration, shock, and tilt (but not rotation) [14].

Fall incidents are increasing; according to the World Health Organization, fatal falls are the second leading cause of accidental death worldwide [15]. Therefore, fall detection systems (FDSs) have attracted much attention in recent decades. According to two most recent survey articles [16,17] on the latest state-of-the-art FDSs, the systems can be classified into three types based on the sensing technologies used. One type uses vision- or sound-based sensors, such as cameras or microphones, to record fall incidents. Another type uses ambient environmental sensors that measure data such as radio frequency identification (RFID) waves, vibrations, pressure changes, or ultrasound, to detect the environmental changes occurred simultaneously with falls. These two types of FDSs are placed in environments in which people tend to fall; therefore, the detections are limited in the fields that have these types of sensors installed. The remaining type of FDSs comprises wearable sensors that detect changes in physical movements or physiological indicators after falls. The advantage of such FDSs is that they can function anywhere and are not limited to certain environments. Almost every wearable FDS uses an accelerometer, and some integrate the accelerometer with other types of sensors, such as gyroscopes [18,19,20,21] or heart rate sensors [22].

However, FDSs can only detect fall accidents that occurred. Although fall detection can help expedite rescue and treatment, it cannot prevent individuals from being injured. Thus, a fall prevention system would be beneficial. Of all hospital fall incidents, 60–70% were bed falls [23]. Bed falls can be sufficiently severe to result in death [24]. Patients recovering from anesthesia after surgery, patients spending long periods confined to a bed, and older adults with motion impairment are highly likely to fall if they attempt to get out of bed without the assistance of caregivers. Although efforts have been made to develop and install bed alarm systems, increased alarm use in urban hospitals was reported to have no statistically or clinically significant effects on fall-related events [25], possibly because bed alarm systems were not triggered in real-time and left not enough time for caregivers to intervene before the patient falls out of bed. Moreover, bed rails may provide additional safety; however, their success rate in preventing bed falls was reported to be low [26]. Bed rails may be able to stop sleeping or unconscious patients from tumbling out of bed; however, conscious individuals with motion impairment might attempt to put down the bed rails and get out of bed. Thus, bed rails are unable to prevent these individuals from bed falls. Because of the lack of adequate patient monitoring and alarm systems, bed fall accidents are common among elderly individuals and patients [27]. Patients with terminally illness in hospitals or at home are also at high risk of such accidents [28].

Fall prediction systems could depend on various indicators. Jähne-Raden et al. [29] developed a system, INBED, for bed exit detection and fall prevention in a geriatric ward by detecting the wakefulness of patients in sleep before their intentions to get out of beds. The system involves a wearable sensor with an inertial measurement unit (IMU), including an accelerometer and a gyroscope, that is attached to a patient’s thigh. The sensor detects leg movements that indicate a change in wakefulness. When the sensor detects wakefulness, it triggers an alarm to ensure immediate assistance for the patient, thus the possible subsequent falls may be prevented. However, this system works for only those patients who are expected to be asleep while in bed; using it for patients who need to lie down only because of motion impairment would result in many false alarms.

INBED system was limited because the fall risk indicator it measured was highly specialized for a certain type of patients. A more common behavior that precedes bed falls in a wide variety of patients is sitting up. Therefore, postural changes from lying down to sitting up can be used as a superior indicator of bed fall risk. Accordingly, timely detection of their attempt to sit up and get out of bed and immediate notification of hospital staff or home caregivers can prevent bed falls, as illustrated in Figure 1.

In this study, an accelerometer-based inertial motion/tilt-sensing IoT module was designed and implemented to detect the postural changes (i.e., from lying down to sitting up) of bedridden patients who are at high risk of falls [30]. The rest of this paper is organized as follows. Section 2 provides details of the design and implementation of the proposed IoT module, namely, Bluetooth low-energy (BLE)-enabled Accelerometer-based Sensing In a Chip-packaging (BASIC) module; the section also describes the posture change detection system. Section 3 presents the verification of the accuracy of the BASIC module in tilt angle detection and the application of the module in bed fall prevention. Moreover, Section 4 discusses the significance of this study and future research directions. Finally, Section 5 provides the conclusions.

## 2. Materials and Methods

This study designed and implemented a BASIC module for detecting the postural changes (i.e., from lying down to sitting up) of individuals at high risk of fall accidents. This IoT module contains an integrated data processing algorithm to offload the burden of computing tilt angle information from raw acceleration data on the application side. With an appropriate configuration (i.e., IoT module orientation when installed), the proposed module can simply assess whether the computed tilt angle (between the individual’s chest and bed surface) exceeds a predefined threshold. If the angle exceeds the threshold, then the individual is considered to be sitting up with the intention of getting out of bed. The sensing and transmission of the tilt angle information by the IoT module obviate the need for bed fall prevention applications to receive and convert raw acceleration data into tilt angles for posture change detection. Thus, the application side has more time and space for other tasks.

### 2.1. Design and Implementation of the BASIC Module

Figure 2 shows the architecture of the BASIC module. The operating principle of the module is to integrate an accelerometer with a BLE module and apply the BLE’s system-on-chip (SoC) to acquire sensing data from the accelerometer and convert them into tilt angles. The patented (TW Patent No. I456202) low-complexity CORDIC-based tilt-sensing algorithm [31] executes data conversion process. The SoC acquires acceleration data from the accelerometer through the I^2^C interface and converts them into tilt angle information directly within the chip. This process is feasible because the tilt-sensing algorithm simply uses logical shift and integer addition/subtraction operations iteratively to achieve highly accurate results. This algorithm enables performing the data conversion process within the BLE SoC along with the original firmware, which handles all BLE communication protocols. Unlike the wearable sensor described in [29], which required a separate processor, IMU, and wireless communication module (Zigbee, 802.15.4), the BASIC module obviates the need for an extra microcontroller for data collection and conversion into tilt angle information, thus, its cost and power consumption can be further reduced. The BLE unit used in the BASIC module is nRF52832 (NORDIC Semiconductor, Oslo, Norway), a Class 2, BT5.0 BLE module with a built-in antenna.

The accelerometer integrated into the module is MIS2DH (STMicroelectronics, Geneva, Switzerland). This accelerometer is an ultra-low-power three-axis sensor specifically designed for medical applications and classified by the U.S. Food and Drug Administration as Class III device. It features user-selectable ±2/±4/±8/±16 g sensing ranges at a 10-bit digital data resolution in normal mode and 12-bit resolution in high-resolution mode. Therefore, it can achieve a fine sensitivity level of 1 mg (1 mg = 2^−10^ g; i.e., 1/1024 g, where g is the gravitational force) in high-resolution mode in the ±2 g sensing range.

Figure 3 depicts a prototype of the BASIC module. Because of its low power requirements, a single CR2032 button battery can power the module for more than 12 h even with continuous BLE data transmission. If the BASIC module transmit data only when the measured tilt angles exceed certain thresholds, it can even work for several days. Notably, because this prototype was mainly designed for research purposes, its size was not optimized. The proposed module measures 60 mm × 29 mm, which can be further reduced using a simplified design involving a stackable structure (battery and printed circuit board (PCB) module); it can even be further miniaturized using system-in-chip packaging techniques.

### 2.2. Over-the-Air Feature Configuration of the BASIC Module

The BASIC module has several features that should be appropriately configured for specific applications, for example, the installation orientation, accelerometer sensing range and sampling frequency selection, coordinate system (Cartesian or spherical coordinate system) representation, and data output modes should be configured. All these configurable features can be set wirelessly through the so-called over the air configuration system on the application side before the application core is executed. Table 1 lists the module features.

As listed in Table 1, the orientation feature represents the orientation of the module installed into an application. It can be perfectly represented by selecting the axis and direction (i.e., X, −X, Y, −Y, Z, or −Z) of the on-module accelerometer, which is aligned along the gravitational force direction in the initial state. By setting the coordinates, users can choose the coordinate system for the conversion and transmission of the tilt angles. The actual meaning of the converted angles also depends on the orientation of the installed module. As illustrated in Figure 4a, if the module orientation is set to −Z, then the angle between the measured gravity vector ($\vec{g}$) and *z*-axis (i.e., φ), the angle between the *x*-axis and projected gravity vector ($\vec{g}$) on the X–Y plane (i.e., θ), and the length of the measured gravity vector (i.e., ρ) are generated if a spherical coordinate system is selected. By contrast, if a Cartesian coordinate system is selected, then the angle between the *x*-axis and horizontal plane (which is perpendicular to the measured gravity vector $\vec{g}$; θ), that between the *y*-axis and the horizontal plane (Ψ), and that between the measured gravity vector ($\vec{g}$) and the *z*-axis (Φ) are generated, as illustrated in Figure 4b.

The tilt angles θ, Ψ, and Φ in the Cartesian coordinate system and θ and φ in the Spherical coordinate system can be calculated using Equations (1) and (2), respectively. The tilt angles in the Cartesian coordinate system are expressed as: (1)$$\theta =\tan^{-1}\frac{A_{x}}{\sqrt{A_{y}^{2}+A_{z}^{2}}}\;\;\Psi =\tan^{-1}\frac{A_{y}}{\sqrt{A_{x}^{2}+A_{z}^{2}}}\;\;\Phi =\frac{\pi }{2}-\tan^{-1}\frac{A_{y}}{\sqrt{A_{x}^{2}+A_{z}^{2}}}\;$$

The tilt angles in the Spherical coordinate system are expressed as:(2)θ=tan−1(AyAx)+σ·λ·πφ=π2−tan−1AyAx2+Az2 
where *A_x_*, *A_y_*, and *A_z_* are the acceleration data for the X-, Y-, and Z-axes. The term σ·λ·π in the formula for the tilt angle θ in the Spherical coordinate system is used to convert θ into the range [−π, π] from the original range [−π/2, π/2] resulting from the tan^−1^ function. Here, *σ* = 0 if *A_x_* ≧ 0; *σ* = 1 otherwise. In addition, λ = 1 if *A_y_* ≧ 0; λ = −1 otherwise.

The sensing range is set to configure the acceleration range measurable by the sensor, and ODR represents the output data packet rate generated by the BASIC module. Although the ODR of the accelerometer can exceed 100 Hz, that of the module is limited to 100 Hz because of the limited bandwidth of BLE modules. Output mode is used to configure the behavior of how the output data packets of the module will be transmitted. If the real-time mode is selected, the module transmits the data packet immediately after it is generated by the accelerometer, and the tilt angles are calculated for every data packet. In burst mode, the module transmits multiple data packets (three packets in our implementation) on a single data transmission. In auto mode, the BASIC module automatically is set to operate in a suitable output data packet mode according to the ODR setting. According to data obtained from testing processes, the BLE module can only transmit 60 bytes of data every 30 ms without data packet loss on the receiving side. Therefore, in a data packet with a maximum size of 20 bytes, the 100 Hz output data packet rate can be achieved only in burst mode. This is because as at the 100 Hz output data rate, the module generates up to 20 bytes of data packet every 10 ms; thus, the cumulative quantity of data to be transmitted every 30 ms is 60 bytes, which can be safely received on the receiving side without packet loss. In Table 1, the underlined options indicate the default values.

Although Equations (1) and (2) are simple and can be found online, the division operations, square roots, and inverse trigonometric functions require floating-point arithmetic. The processor requires built-in hardware support for floating-point operations to convert the accelerometer data into tilt angles accordingly; therefore, the processor may still require considerable time and instructional code to perform the computational tasks. Therefore, a CORDIC-based tilt-sensing algorithm is implemented to avoid these difficulties.

### 2.3. Data Packet Deign of the BASIC Module

As mentioned, to achieve a maximum ODR at 100 Hz, the size of each data packet should be limited to 20 bytes. Accordingly, the data packet design of the proposed BASIC module is presented in Figure 5. In this design, each data packet contains 6 bytes of raw acceleration data measured (2 bytes each for the X-, Y-, and Z-axes); 6 bytes of converted tilt angle information (2 bytes for tilt angles and 1 byte for vector length (i.e., θ, φ, and ρ) in the spherical coordinate system; 3 bytes for tilt angles (i.e., θ,Ψ, and Φ) in the Cartesian coordinate system); and 4 bytes of data for the header, tail, counter, and checksum. Thus, this data packet contains a total of 16 bytes.

### 2.4. Calibration of the BASIC Module

To improve the accuracy of the acceleration and tilt angles sensed by the module, the module should be calibrated. The module’s accuracy might be affected by multiple error sources, primarily errors from the accelerometer itself and those generated due to the attachment (placement and soldering) of the accelerometer to the module.

Each manufactured accelerometer is typically calibrated with respect to its reference sensitivity, frequency response, resonant frequency, and time constant; nevertheless, the output bias or zero-g offset and gain may differ considerably between devices and modules. In particular, if the sensor chip (i.e., the accelerometer) is not perfectly flushed with the PCB module during the mounting process, then differences in the zero-g level on the module and slight tilt variances between the sensor device and module might occur. Furthermore, in the event of any slight misalignment of the sensor chip on the module, the X-, Y-, and Z-axes directions would differ between the accelerometer and module. Therefore, the module must be calibrated using acceleration data, which can be achieved using the following equation:Calibrated measurement (in unit of g) = (RAW_data_from_Accelerometer-offset)/gain(3)

In this study, calibrations were performed on the X-, Y-, and Z-axes of the module separately on a controllable three-axis rotation platform (with a tri-axis step motor controller, TL-3T, purchased from Tanlian E-O Co. Ltd., Tao-Yuan, Taiwan). The calibration processes are described as follows (Figure 6):
The rotation platform was perfectly aligned with the horizontal plane, and the module was installed on the surface such that the edge with golden pads was in alignment with the platform’s edge, as shown in Figure 6a. Thus, the *z*-axis of the module (the orientation of the module shown in Figure 6b) could be aligned with the gravitational force direction. The terms X_offset1_, Y_offset1_, and Z_1g_ represent the acceleration data.The platform was rotated on the *y*-axis by 90° so that the *x*-axis of the module could be aligned with the gravitational force direction, as shown in Figure 6b. The terms X_1g_, Y_offset2_, and Z_offset1_ represent the acceleration data.From step 1, the platform was rotated on the *x*-axis by 90° so that the *y*-axis of the module can be aligned with the gravitational force direction, as shown in Figure 6c. The terms X_offset2_, Y_1g_, and Z_offset2_ represent the acceleration data.

All acceleration readings were taken from 100 consecutively sampled data, on average, while the platform remained fixed. In step 1, the ideal readings of the acceleration data should be Acc_X = Acc_Y = 0 and Acc_Z = 1 g; in step 2, the ideal readings should be Acc_X = 1 g and Acc_Y = Acc_Z = 0; and in step 3, the ideal readings in the X-, Y-, and Z-axes should be 0, 1 g, and 0, respectively. Therefore, the actual 1-g readings in the X-, Y-, and Z-axes obtained in steps 2, 3, and 1, respectively, could serve as the gains of the respective axes for calibration. The zero-g level offset on each axis could be derived by calculating the averages of the readings obtained in steps 1, 2, and 3 under the assumption that they would be 0 s ideally.

The acceleration readings obtained from the BASIC module according to the aforementioned calibration steps are summarized in Figure 7.

Therefore, the acceleration readings for the BASIC module can be calibrated using Equations (2)–(4).
(4)$$\mathrm{Acc}\_\mathrm{CalX}\;(\mathrm{in}\;\mathrm{unit}\;\mathrm{of}\;g)=(\mathrm{Acc}\_\mathrm{RawX}-\frac{\left(X_{\mathrm{offset1}}+X_{\mathrm{offset2}}\right)}{2})/X_{1g},$$
(5)$$\mathrm{Acc}\_\mathrm{CalY}\;(\mathrm{in}\;\mathrm{unit}\;\mathrm{of}\;g)=(\mathrm{Acc}\_\mathrm{RawY}-\frac{\left(Y_{\mathrm{offset1}}+Y_{\mathrm{offset2}}\right)}{2})/Y_{1g},$$
(6)$$\mathrm{Acc}\_\mathrm{CalZ}\;(\mathrm{in}\;\mathrm{unit}\;\mathrm{of}\;g)=(\mathrm{Acc}\_\mathrm{RawZ}-\frac{\left(Z_{\mathrm{offset1}}+Z_{\mathrm{offset2}}\right)}{2})/Z_{1g},$$
where Acc_RawX, Acc_Raw_Y, and Acc_RawZ represent the original acceleration readings acquired directly from the accelerometer and Acc_CalX, Acc_CalY, and Acc_CalZ represent the acceleration data calibrated for the module. These calibrated parameters (i.e., X_offset1_, X_offset2_, X_1g_, Y_offset1_, Y_offset2_, Y_1g_, Z_offset1_, Z_offset2_, and Z_1g_) are stored in the nonvolatile memory of each module, and only the calibrated data are transmitted by the BASIC module. The tilt angle conversion algorithm executed inside the BASIC module also uses the calibrated acceleration data to generate more accurate angle information.

### 2.5. Posture Change Detection Methodology

As proposed in [30], an individual attempting to get out of bed changes their posture from lying down to sitting up. For people at high risks of bed falls, such posture change can be considered to indicate their intention to get out of bed. Therefore, if such changes can be detected and transmitted to medical-care personnel or caretakers immediately, bed fall accidents can be prevented.

Accordingly, for posture change detection, the application with the BASIC module can be attached to the chest of an individual, as illustrated in Figure 8a, where the *x*-axis of the module points toward the head. When the individual is lying on the bed, the module is almost parallel to the horizontal plane (i.e., the bed surface). Therefore, the tilt angle θ between the *x*-axis of the module and the horizontal plane is below the prespecified angle threshold $\theta_{thr2}$. When the individual attempts to sit up on the bed from the lying posture, θ exceeds the other prespecified angle threshold $\theta_{thr1}$. Thus, through the continuous monitoring of θ by the BASIC module, the individual’s posture change can be detected promptly, as shown in Figure 8b. The condition of this posture change can be expressed as follows in Equation (5):(7)$$\theta \geq \theta_{thr1},\;\mathrm{alarm}\;\mathrm{goes}\;\mathrm{off}\;\theta <\theta_{thr2},\;\text{dis-alarm}$$

When θ exceeds $\theta_{thr1}$, an alarm (a light and sound alarm) is sent to the caretaker to prompt them to provide immediate assistance, which can prevent fall accidents. By contrast, when θ decreases below $\theta_{thr2}$, the alarm is turned off.

## 3. Results

This study verified the accuracy of the data, especially the tilt angles, measured by the BASIC module to demonstrate the reliability of the module for real-time applications. The application of the module for the bed fall prevention was also explored.

### 3.1. Subsection Tilt Angle Verification of the BASIC Module

To evaluate the accuracy of the tilt angles measured by the BASIC module, the same rotation platform as that used in the calibration processes was used. As described in step 1 of the calibration processes, the BASIC module was initially installed horizontally on the platform so that its X–Y plane was parallel to the platform’s surface. In addition, the *z*-axis of the module was aligned with the gravitational force direction. At this position, the *x*-axis was aligned with the central line of the platform and pointed toward the ring of the rotation angle meter, as indicated in Figure 9.

From the initial position, the platform rotated 360° clockwise at 15° steps about its center axis (i.e., *x*-axis of the BASIC module). The platform was maintained at each rotation step for more than 4 s to ensure that at least 100 consecutive acceleration and tilt angle data points could be recorded (e.g., by setting the sampling rate of the BASIC module to 25 Hz) and averaged. Acceleration data for the X-, Y-, and Z-axes were used to calculate the tilt angles *θ*, Ψ, and Φ (Cartesian coordinate system) and *θ* and *φ* (Spherical coordinate system) to evaluate the accuracy of tilt angle detection and calibrate the module. Figure 10a shows a plot of the theoretical values of *θ*, Ψ, and Φ, where −π/2 ≦ *θ*, Ψ ≦ π/2, and 0 ≦ Φ ≦ π. Figure 10b presents a plot of the angle errors (for *θ*, Ψ, and Φ) obtained with the differences between the theoretical values and calculated values using the raw acceleration data before calibration. Figure 10c presents a plot of the angle errors calculated using acceleration data after calibration. Figure 10d presents a plot of the theoretical values of *θ* and *φ*, where *θ* = π/2 and 0 ≦ *φ* ≦ π. In this verification process, because the platform was rotating about the *x*-axis, the projective gravity forces on the X–Y plane of the accelerometer were always aligned with the *y*-axis. Therefore, the theoretical values of the angle *θ* were π/2 (i.e., 90°) for all rotation steps. Figure 10e presents a plot of the calculated angle errors (*θ* and *φ*) with the theoretical values obtained using the raw acceleration data before calibration. Figure 10f shows a plot of the angle errors calculated using data after calibration. The horizontal axes of all plots in Figure 10. indicate the rotation angles of the testing platform. The vertical axes indicate the angle values in Figure 10a,d and the errors of the calculated angles in Figure 10b,c,e,f. The errors of the tilt angles calculated using acceleration data calibrated through the methods described in Section 2.4, Figure 10c,f, are significantly lower than those calculated using raw acceleration data (without calibration; Figure 10b,e).

The values of the angle errors are listed in Figure 11. For the calibrated data, the maximum tilt angle errors are< 0.8° except for those observed at the 0°, 180°, and 360° rotational steps in the Spherical coordinate system. The high errors of the tilt angle θ at those positions result from *A_x_* being approximately zero. When the tilt angles θ are calculated with Equation (2) for those positions, the error can be considerable with even tiny noise values on *A_x_*. Therefore, otherwise negligible noise in the *A_x_* value results in a large θ error.

These results verify that the calibration method described in Section 2.4 considerably improves the accuracy of the tilt angle calculation.

### 3.2. Development and Implementation for Bed Fall Prevention Application

After verifying the accuracy of the tilt angles obtained by the BASIC module, this study implemented it in an application for bed fall prevention. To demonstrate the practical applicability of the proposed module, the application was tested in a real scenario. In this scenario, a male individual lay on a bed with the BASIC module mounted on his chest, as illustrated in Figure 8a. After lying on the bed in the supine position, he sat up with the intention to get out of bed. The application detected his postural change as he sat up and sent an alarm, as shown in Figure 12a. Subsequently, he stood on the ground and took several steps. After that, he lay down again on the bed in the supine position, and the alarm was disabled, as in Figure 12b. The above sequence of activities, i.e., from lying, sitting, standing up and walking, going back to the bed side, and then lying down again, is defined as pattern of activity, Activity I, and will be referred in the following discussion.

The application was designed on an Android-based tablet PC. The BASIC module mounted on the individual’s chest was preconfigured as follows. Cartesian coordinates were used, the *z*-axis was aligned with the gravitational force direction, the sensing range was ± 2 g, the sampling rate was 100 Hz, $\theta_{thr1}$ was set to 30°, $\theta_{thr2}$ was set to 25°, and the data output mode was “Auto.” In this demonstration, the application received the data including the tilt angles, from the BASIC module through Bluetooth, continuously and executed the posture change detection algorithm, as described in Section 2.5, in the application. Specifically, the application checked whether the tilt angle θ (i.e., the tilt angle between the *x*-axis of the module [accelerometer] and the bed surface [horizontal plane]) exceeded $\theta_{thr1}$, indicating that the patient had sat up. If so, the application would trigger a warning alarm and turn on a light bulb until the received tilt angle θ was less than $\theta_{thr2}$, indicating that the patient had lain back down. Because the BASIC module performs the tilt angle conversions, the application simply implements the postural change algorithm and activates or disables the alarm without computing the tilt angles from the received acceleration data.

Figure 13 illustrates the detected tilt angle θ versus time during the activity, Activity I, shown in Figure 2. In the figure, the red horizontal line represents the threshold value of the tilt angle θ $(\theta_{thr1});$ θ values above this threshold trigger the alarm. The green horizontal line represents the threshold value of the tilt angle θ ($\theta_{thr2}$); θ values below this threshold disable the alarm. The horizontal axis indicates time. The values of the tilt angle θ converted from the data in each packet from the BASIC module were plotted. The ODR was set to 100 Hz, and thus the time between two consecutive data points was 10 ms.

To further verify the applicability of the proposed postural change detection algorithm, the tester performed additional actions (Activity II). The tester lay on the bed in the supine position and then began to roll leftward. The tester continued rolling until he was in the prone position and continued to face to his right. From this position, the tester got out of bed and stood on the floor. As in Figure 13, the $\theta_{thr1}$ and $\theta_{thr2}$ threshold values are indicated by horizontal lines in Figure 14. The tilt angle θ was less than $\theta_{thr1}$ while the tester rolled; therefore, the alarm was not triggered. However, when the tester rose to get out of bed, the tilt angle θ exceeded $\theta_{thr1}$, and the alarm was triggered. The tilt angle θ versus time is plotted in Figure 14.

## 4. Discussion

This paper presents the design, implementation, and validation of an IoT module, namely, the BASIC module, for inertial motion and tilt sensing. The concept underlying this module is to detect the posture change (from lying down to sitting up) of an individual at a high risk of bed falls; this posture change is an indicator of the individual’s attempt to get out of bed. Accordingly, this module is a response to calls for state-of-the-art IoT applications for nursing care, as specified in [32]. The proposed BASIC module and its application can be implemented in mobile devices for bed fall prevention at home or other care settings. To deploy this application to multiple patients in hospital wards, care centers, and clinical institutes, a gateway device would be required to collect data from the multiple BASIC modules applied to various patient and transmit the information through a wired or wireless network to the nursing station. The BASIC module uses an nRF52832 BLE device, a Bluetooth Class 2 device with a maximal communication range of approximately 10 m. A gateway device, installed in the room, collects Bluetooth signals from different BASIC modules and transmits them to the nursing station, allowing nurses to monitor patients more than 10 m from their station and receive warning alarms.

The proposed BASIC module has not yet been miniaturized and was implemented only for application validation. For future applications, the size of the BASIC module can be reduced, and a different battery can be used. The CR2032 button battery can be replaced with a rechargeable battery to improve usability and operating duration. The miniaturized module can be packaged into any form of carrier, for example, it can be integrated with a bandage for easy attachment to the body or can be fixed to a pin, a button, or any design that can facilitate its attachment to clothes and enable. The BASIC module can be turned into a wearable device, thereby enabling the system to be used for real-time medical care. In this proof-of-concept demonstration for bed fall prevention, the application performed postural change detection based on data streamed from the BASIC module. In a future device, posture detection can be performed in the BASIC module itself because the algorithm is simple and can be implemented by the BLE module’s SoC. This could further reduce power consumption because the BLE module would not need to continually transmit data packets. Instead, data would only be transmitted when the alarm is triggered or disengaged. The data sampling rate could also be lowered from 100 to 25 Hz to further reduce power consumption. Past research has reported that 20 Hz is sufficient for the successful identification of activities [33].

## 5. Conclusions

This study developed an accelerometer-based, low-cost, real-time motion- and tilt-sensing IoT module, namely, a BASIC module. The study also developed an application involving this module for detecting an individual’s postural change (from lying down to sitting up) as an indicator of the individual’s intention to get out of bed; the application was determined to be suitable for bed fall prevention. The model includes a patented CORDIC-based tilt-sensing algorithm integrated directly into the SoC of the BLE unit, thus rendering the module unique and considerably reducing application-side tasks. In the future, the application can be implemented in hospitals or other institutions with a gateway system designed to collect data transmitted to nursing station by several BASIC modules in a ward. The BASIC module can be further miniaturized and packaged into a wearable device, and the power consumption can be further reduced. This can help save medical resources and prevent injury, or even mortality, caused by bed fall accidents.

## 6. Patents

The low-complexity CORDIC-based tilt sensing algorithm implemented in the BASIC module reported in this article is patented under TW Patent No. I456202.

## Figures and Tables

**Figure 1 biosensors-11-00428-f001:**
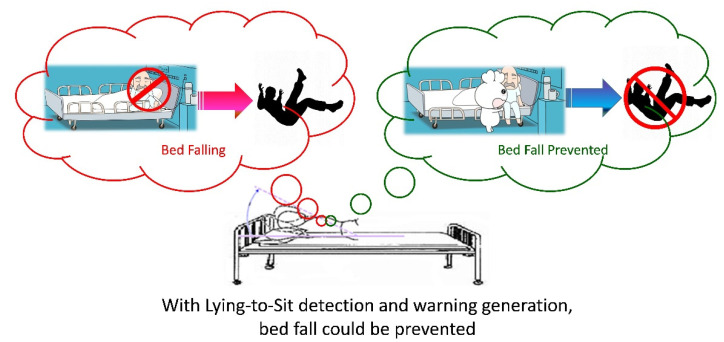
Bed fall prevention through posture change (from lying down to sitting up) detection and warning.

**Figure 2 biosensors-11-00428-f002:**
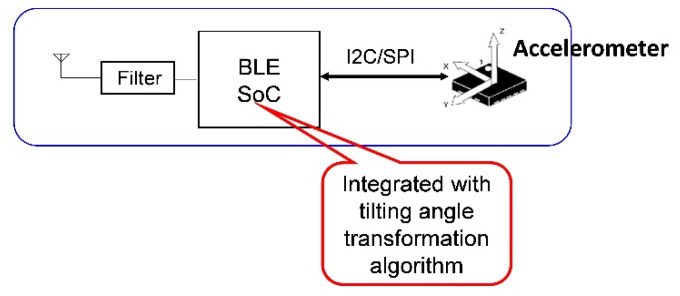
Architecture of the BASIC module.

**Figure 3 biosensors-11-00428-f003:**
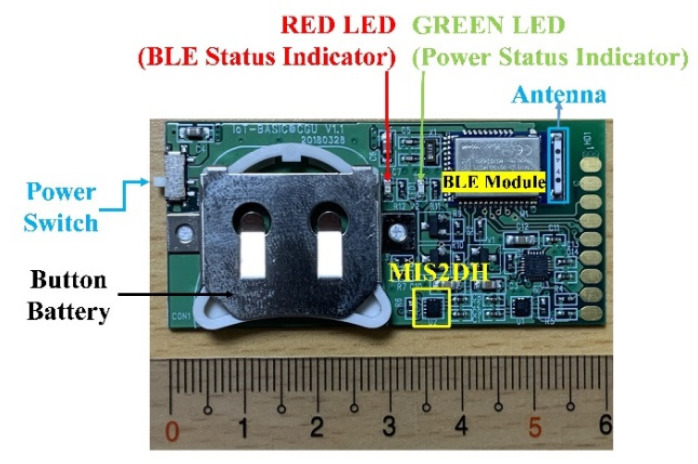
Prototype of the BASIC module.

**Figure 4 biosensors-11-00428-f004:**
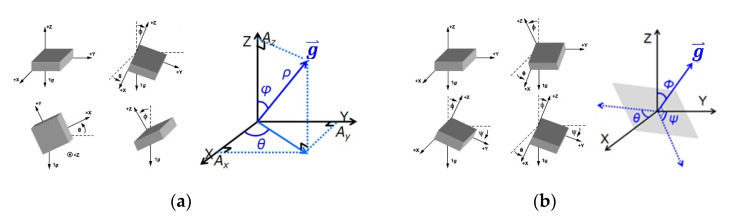
Coordinate systems with their defined tilt angles. (**a**) Spherical Coordinates, (**b**) Cartesian Coordinates.

**Figure 5 biosensors-11-00428-f005:**
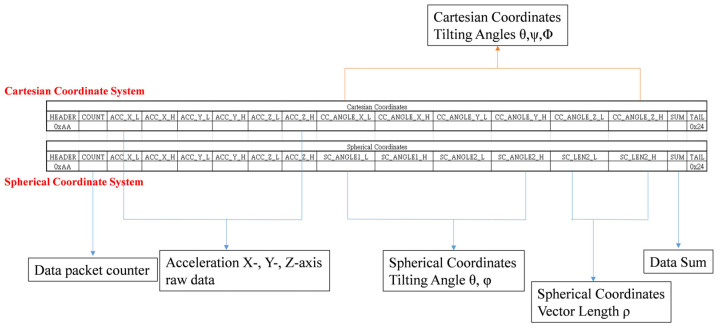
Data packet format of the BASIC module.

**Figure 6 biosensors-11-00428-f006:**
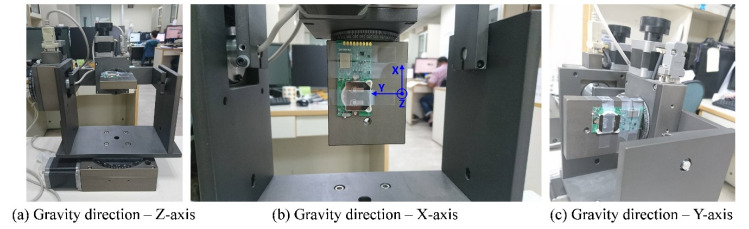
BASIC module calibration processes: (**a**) gravitational force direction—*z*-axis; (**b**) gravitational force direction—*x*-axis; (**c**) gravitational force direction—*y*-axis.

**Figure 7 biosensors-11-00428-f007:**
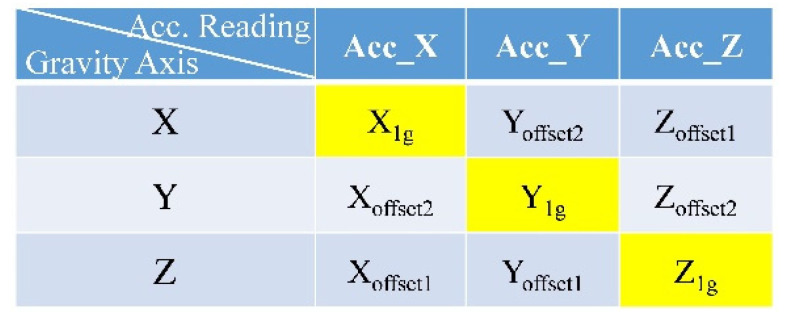
Acceleration readings obtained from the BASIC module for calibration.

**Figure 8 biosensors-11-00428-f008:**
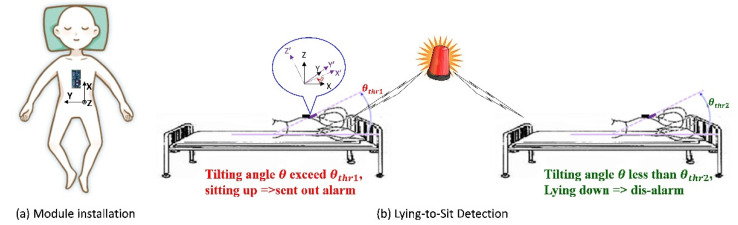
(**a**) BASIC module installation. (**b**) Detection of posture change (from lying down to sitting up) and alarm.

**Figure 9 biosensors-11-00428-f009:**
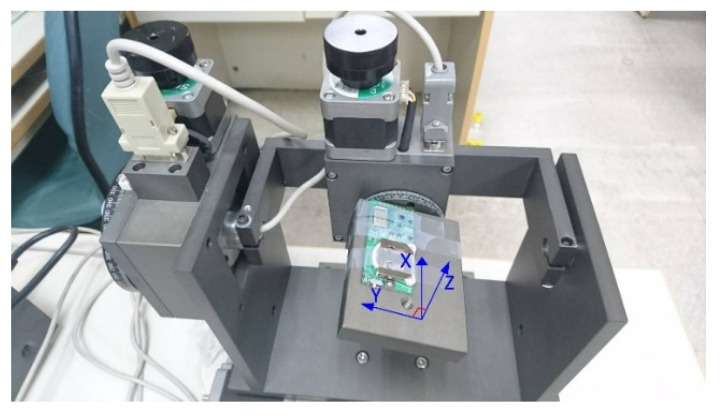
Tilt angle verification (rotating on accelerometer *x*-axis).

**Figure 10 biosensors-11-00428-f010:**
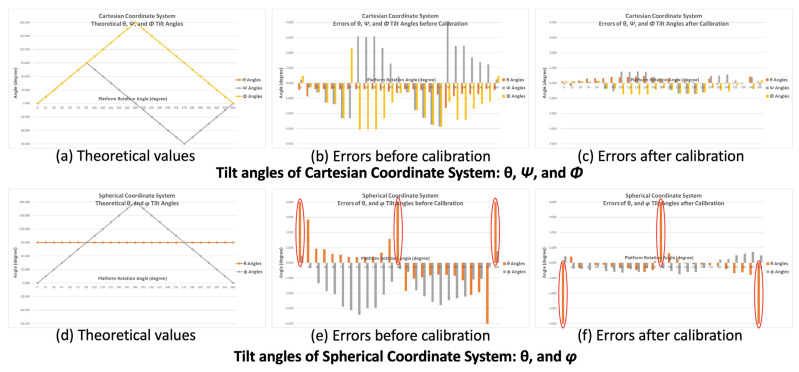
Plots of the tilt angles. *θ*, Ψ, and Φ in the Cartesian coordinate system. (**a**) Theoretical values. (**b**) Errors of angles calculated from raw acceleration data before calibration. (**c**) Errors of angles calculated from calibrated acceleration data. *θ* and *φ* in the Spherical coordinate system. (**d**) Theoretical values. (**e**) Errors of angles calculated from raw acceleration data before calibration. (**f**) Errors of angles calculated from calibrated acceleration data.

**Figure 11 biosensors-11-00428-f011:**
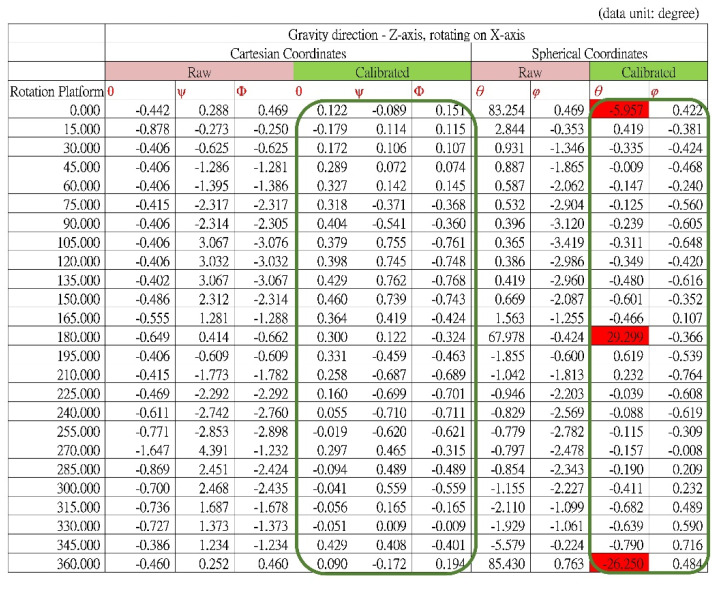
Results of tilt angle verification.

**Figure 12 biosensors-11-00428-f012:**
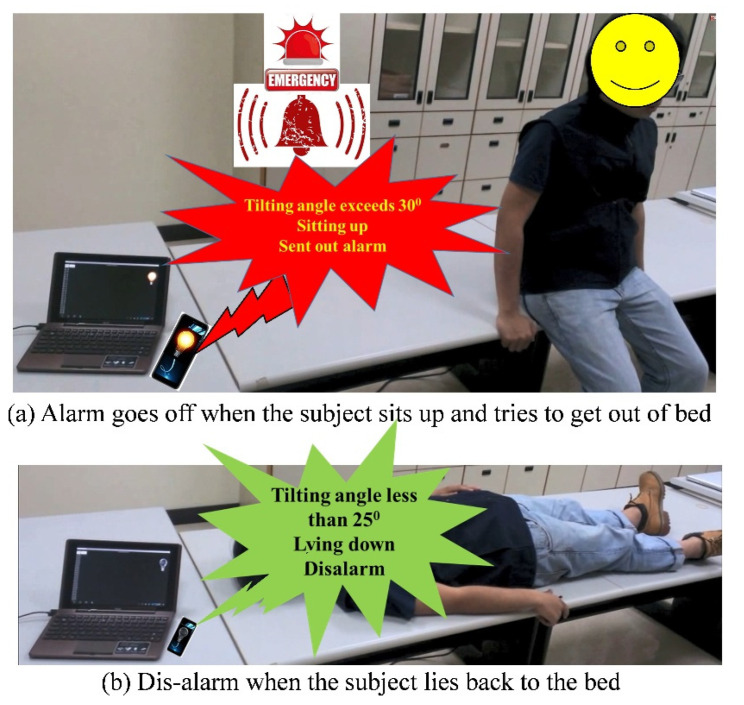
Bed fall prevention application scenario. (**a**) Alarm triggered when the individual sits up and attempts to get out of bed. (**b**) Alarm is disabled when the individual lies back on the bed.

**Figure 13 biosensors-11-00428-f013:**
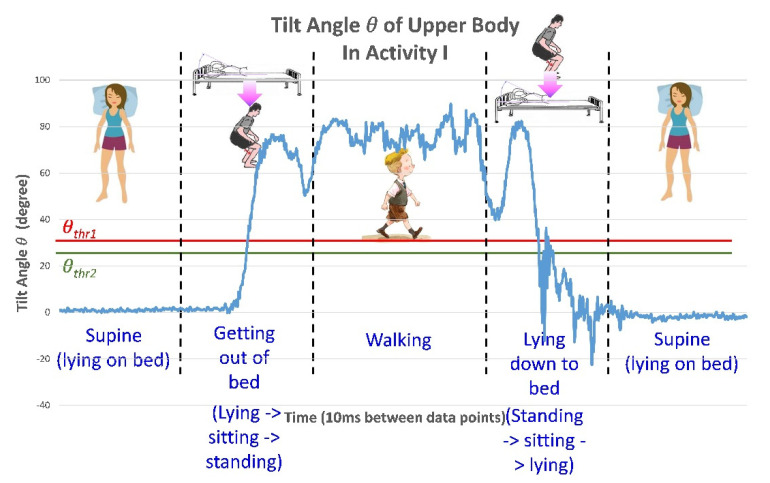
The detected tilt angle *θ* during postural change in Activity I. The tester sat up, stood up, took several steps, and then lay down again in the supine position (Activity I).

**Figure 14 biosensors-11-00428-f014:**
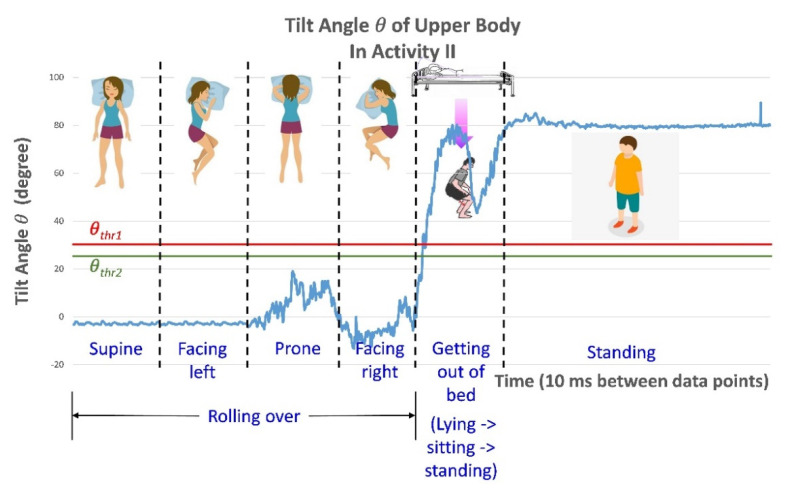
The detected tilt angle *θ* in the defined activity pattern, Activity II.

**Table 1 biosensors-11-00428-t001:** Configurable features of the BASIC module.

Features	Options	Descriptions
Orientation	X, −X, Y, −Y, Z, or −Z	Axis and direction of the accelerometer which is on the same direction of gravity force
Coordinates	Cartesian or Spherical	Which coordinate system of the tilting angles will be generated
Sensing Range	±2 g, ±4 g, ±8 g, or ± 16 g	Acceleration sensing range of the accelerometer
Output Data Rate (ODR)	1, 10, 25, 50 or 100 Hz	Output data rate of the module, i.e., accelerometer
Output Mode	Real-time, Burst, or Auto	The data packet output mode

## Data Availability

The data presented in this study are available on request from the corresponding author.

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
