# Peer review of "Design and Implementation of a Wearable Accelerometer-Based Motion/Tilt Sensing Internet of Things Module and Its Application to Bed Fall Prevention"

_biosensors, 2021, doi:10.3390/bios11110428_

Round 1

Reviewer 1 Report

In this paper, the authors designed and implemented an Internet-of-Things module for detecting the posture change (from lying down to sitting up), which shows the capabilities for the motion/tilt sensing and its application to bed fall prevention of individuals. This is a carefully done study, and these findings are of considerable interest. Therefore, I think this manuscript can be published after minor revise. And I have some specific comments on the technical part that might help further improve the work.

1) For the accelerometer, there are different operating modes, sensing ranges, and output data rates, how can the modes be changed? Or is there only one mode being used?

2) Line 307, tilt angle data points could be recorded (e.g., by setting the sampling rate of the BASIC module to 20 Hz), but as can be seen in Table 1, the accelerometer does not have an ODR of 20 Hz?

3) In Figure 10, the unit of the angle is missing.

4) About abbreviations in the text, it is recommended that the full name be given at the first occurrence.

5) The format of the references is recommended to be adjusted and meet the requirements of the journal.

Author Response

  • Thanks for the reviewer’s comments and positive affirmation. The detailed responses to the other comments are listed below.
  • Comment 1: Extensive editing of English language and style required

[Response:] The revised manuscript has gone through the extensive English editing service. Please see the attached certificate in the cover letter.

  • Comment 2: For the accelerometer, there are different operating modes, sensing ranges, and output data rates, how can the modes be changed? Or is there only one mode being used?

[Response:] The basic operation modes of accelerometer, such as: sensing ranges, and output data rates can be changed from the applications interacting with the BASIC module by the pre-defined configuration commands of the Over-the-Air Feature Configuration. Please refer to section 2.2 for details. As for the bed fall prevention application demonstrated in this work, the sensing range was set to ±2g, with the out data rate (sampling rate) set to 100Hz on the BASIC module, which was also described in Sec. 3.2.

  • Comment 3: Line 307, tilt angle data points could be recorded (e.g., by setting the sampling rate of the BASIC module to 20 Hz), but as can be seen in Table 1, the accelerometer does not have an ODR of 20 Hz?

[Response:] Thanks for pointing out the typos. It has been corrected as “more than 4 s to ensure that at least 100 consecutive acceleration and tilt angle data points could be recorded (e.g., by setting the sampling rate of the BASIC module to 25 Hz)”.

  • Comment 4: In Figure 10, the unit of the angle is missing.

[Response:] The data (angles) in Figure 10 (in this revised manuscript, it is now Figure 11) are in degrees. Figure 11 has been modified with the unit of angle, “(data unit: degree)”, added on the upper-right corner of the figure.

  • Comment 5: About abbreviations in the text, it is recommended that the full name be given at the first occurrence.

[Response:] Thanks for pointing out this. The full names of ECG, EMG, EEG, CORDIC, and ODR are given at their first occurrences.

  • Comment 6: The format of the references is recommended to be adjusted and meet the requirements of the journal.

[Response:] Thanks for point out the issue. All the references are carefully checked to meet the requirements of the journal.

Reviewer 2 Report

The authors demonstrated the use of a MIS2DH-accelerometer-based motion sensing device (namely BASIC module) and a patented low-complexity CORDIC-based tilt-sensing algorithm to detect the posture change of a person from lying down to sitting up, which is considered a risk of falling out of the bed. The attraction of this study relies on how to predict falls as well as identify high possibilities of fall accidents, instead of detecting only fall accidents. The manuscript is well written. However, the reviewer recommends this manuscript to be published in other journals instead of Biosensors, because of following reasons: (1) The study doesn’t use or mention bio-related materials, devices, or technologies – it means the study is out of the scope of Biosensors journal. (2) The raw and calibrated data of tilt angles (θ, Ψ, Φ) from the BASIC module should be presented by plots for demonstrating how the coordinate values or the electrical signals received from the module have changed before, during, and after the events of posture changes. Only Figure 10 only shows differences (errors) between the tilt angles. Please refer the presentation of motion data from Sensors 2019, 19(5), 1017; DOI: 10.3390/s19051017. (3) The dataset is quite limited, and the number of trials with different postures has not been considered (e.g., turning left, turning right, facing down, and rolling), and thus, it is insufficient reliable to conclude that the developed module is suitable for bed-fall-prevention applications. (4) The study is lacking considerations about effects of sitting-up velocity and breathing* on (1) signal/value errors and (2) the activation or deactivation time of alarm. *Here, the sensing module was placed on the abdomen of the person, as illustrated in Figure 8.

Author Response

  • Thanks for the reviewer’s comments. The responses to the reviewer’s comments can be found below. Parts of the responses were also added in the revised manuscript.
  • Comment 1: The manuscript is well written. However, the reviewer recommends this manuscript to be published in other journals instead of Biosensors, because of following reasons: (1) The study doesn’t use or mention bio-related materials, devices, or technologies – it means the study is out of the scope of Biosensors journal.

[Response:] Thanks for the review’s positive comment about the written of the original manuscript. Regarding submitting to other journals because that it was out of the scope of Biosensors, we were kind of hesitated for if it was proper to submit for publication in the Biosensors journal at the beginning. However, after we looked at the message from the guest editors of this special issue more carefully, it mentioned “biological or physiological signals from the human body” and also in the listed topics, “wearable sensors” was covered in most of the topics. We believed that the concepts disclosed in this manuscript should within the scope of this special issue in Biosensors journal. Moreover, 3 out of 4 articles published in this special issue at the date of Oct. 1, 2021, were not involving with bio-related sensors. This really encourage us to submit this manuscript to this special issue for publication.

  • Comment 2: The raw and calibrated data of tilt angles (θ, Ψ, Φ) from the BASIC module should be presented by plots.

[Response:] Thanks for the comment. A new figure (Figure 10 in the major revision) showing the data plots for the tilt angles (θ, Ψ, Φ) and (θ,φ) of their theoretical values as well as the errors of the tilt angles using raw and calibrated data has been added in the revised manuscript for better understanding with plots.

  • Comment 3: for demonstrating how the coordinate values or the electrical signals received from the module have changed before, during, and after the events of posture changes. Please refer the presentation of motion data from Sensors 2019, 19(5), 1017; DOI: 10.3390/s19051017.

[Response:] Thanks for the suggestion. The values of the converted tilt angle θ in Cartesian Coordinate System which is used to determine if the subject intended to get out of bed were plotted in Figure 13 of the major revision of this manuscript for the demonstration activity patterns. The kind of activities that the subject was performing was also annotated in the figure.

  • Comment 4: The dataset is quite limited, and the number of trials with different postures has not been considered (e.g., turning left, turning right, facing down, and rolling), and thus, it is insufficient reliable to conclude that the developed module is suitable for bed-fall-prevention applications.

[Response:] Thanks for the comments. As pointing out from the reviewer and also to verify the applicability of the postural change algorithm in Sec. 2.5 to see if the developed module along with the postural change algorithm is suitable for bed-fall prevention applications. We have a subject to lie on the bed in supine position initially. Then the subject started to roll over to his left hand side, continue the rolling over, so the subject would be in facing left, prone (facing down), and facing right positions. Then the subject got out of bed and stand on the floor. The calculated tilt angles θ were plotted as these activities of rolling over and getting out of bed going. From this plot, we can observe that the angles θ never exceeded θthr1, and hence this testing further verified that the BASIC module with the proposed postural change algorithm is suitable for bed-fall prevention.

Reviewer 3 Report

This work describes a developed Bluetooth module to detect the posture change of individuals with a high risk of falls from the beds. The theme of the work is interesting. It is my opinion that this work and the manuscript should be improved: - Some of the references are old and not state-of-the art references. - The conversion procedure of acceleration to tilt angle is a simple procedure, that can be found online, for example, in any Arduino MEMs example. - (1 mg = 2−10 g; i.e., 1/1024 g, where 147 g is the gravitational force) is incoherent. - 100-Hz should be 100 Hz. ±2-g should be ±2 g. - The nRF52832 SoC range (in meters) is not mentioned. What is the average range of the developed device? Is it suitable for nurse application is the nurse is outside the room? - The quality of figures 5 and 10 must be improved. - The tablet PC designed application must be described with detail. - The phrase mentioned "Thus, nurses can monitor and receive 360 alarms in the station whenever any patient in the ward attempts to get out of bed" refers to a goal that not implemented in this work. - The CR2032 battery is not rechargeable. It is feasible to use this module in field without rechargeable power capability. How many times the CR2032 battery must be replaced for a new one?

Author Response

  • Thanks for the reviewer’s comments. The responses to the reviewer’s comments can be found below. Parts of the responses were also added in the revised manuscript.
  • Comment 1: Some of the references are old and not state-of-the art references.

[Response:] Thanks for the valuable comments. The parts which referenced to the old and not state of the art references were revised totally in this major revision. Now, most of the works in introduction were referenced from recent state of the art works.

  • Comment 2: The conversion procedure of acceleration to tilt angle is a simple procedure, that can be found online, for example, in any Arduino MEMs example.

[Response:] Thanks for pointing this out. The mathematical equations for calculating these tilt angles were added in Equation (1) and (2) in Sec. 2.2. Indeed, these equations are quite simple and can be found online easily, however, the calculations of these equations require using floating point to compute division operations and also execute square root as well as inverse trigonometric functions, i.e. tan-1 functions. To be able to convert the measured raw data from accelerometers into tilt angles accurately using these equations, the processor requires build-in hardware supports for floating point operations and may still take considerable amount of time and instruction codes space for the tasks. CORDIC-based tilt-sensing algorithm used integer addition and shifting operations iteratively, so it will be faster and require less code spaces to calculate reasonable accurate tilt angles in lower-end microprocessor IoT module. The implementation of CORDIC-based tilt-sensing algorithm can reduce the time and memory space for the codes required to convert acceleration into tilt angles using mathematical equations. These arguments were also addressed in Sec. 2.2 of this majorly revised manuscript.

  • Comment 3: (1 mg = 2−10 g; i.e., 1/1024 g, where g is the gravitational force) is incoherent. - 100-Hz should be 100 Hz. ±2-g should be ±2 g.

[Response:] Thank you for point out these inconsistency of presentation in the manuscript. They have been fixed and all other similar presentations were also checked to make sure that they are all consistent.

  • Comment 4: The nRF52832 SoC range (in meters) is not mentioned. What is the average range of the developed device? Is it suitable for nurse application is the nurse is outside the room?

[Response:] Thanks for the comments. The nRF52832 is a Class 2 BLE module which has a maximum of 10 m communication range. It might not be suitable for the direct application for the nursing station far away. So a gateway system inside the ward to collect multiple BASIC warning messages and convert them into other wireless or wired signals working for longer range communication. With this gateway device, the bed-fall prevention system will be applicable to the nursing station even far away. This has been added in the Sec. 4, Discussion, for future implementation.

  • Comment 5: The quality of figures 5 and 10 must be improved.

[Response:] It is grateful for the suggestion. The resolution of Figure 5 and Figure 10 (now is Figure 11 in this revision) had been improved in the revised manuscript.

  • Comment 6: The tablet PC designed application must be described with detail.

[Response:] Thank you very much for the valuable comments. The tasks that the tablet PC application software have to complete are explained in Sec. 3.2. in this revised manuscript. With the BASIC module that will be able to convert and transmit the angle or even integrate the detection of posture change of getting out of bed and sending the warning messages only, the tasks of the tablet PC software are quite simple to perform the angles or messages checking only for the alarm triggering or turn-off.

  • Comment 7: The phrase mentioned "Thus, nurses can monitor and receive alarms in the station whenever any patient in the ward attempts to get out of bed" refers to a goal that not implemented in this work.

[Response:] Thank you for the comments. Indeed, the main purpose of this manuscript is to present the design of BASIC module and propose algorithm for bed-fall prevention. The work been done is for proof of concept only. How to complete a full functional system is discussed in Sec. 4, Discussion, and it has been explicitly disclosed in the discussion section.

  • Comment 8: The CR2032 battery is not It is feasible to use this module in field without rechargeable power capability. How many times the CR2032 battery must be replaced for a new one?

[Response:] Thanks for the valuable comments. The battery design for the future commercialized product can be change to other rechargeable battery for the usability improvement. Currently, the battery is able to works for more than 12 hours. However, as in the discussion section, Sec. 4, the power consumption of the BASIC module could be even lowered by setting to lower ODR, such as 25Hz, or even to have the postural change algorithm implemented inside the module, so that the module will not need to continually transmit out the sensed data but will only send the warning message when the intention of getting out of is detected. By this way, the working hours with CR2032 will expect to be last for several days.

Round 2

Reviewer 2 Report

The authors have answered my comments and improved the manuscript significantly. Although there exists a few minor errors that can be corrected during the proof correction stage, the reviewer suggests that the manuscript is acceptable for publication in Biosensors. 

Reviewer 3 Report

The authors improved, in general, the manuscript according to the reviewer comments.

Figure 10 quality must be improved. The manuscript has errors: ex: [Error! Reference source not found.]